# Impact of review method on the conclusions of clinical reviews: A systematic review on dietary interventions in depression as a case in point

Florian Thomas-Odenthal[1][☯]*, Patricio Molero[2][☯], Willem van der Does[1,3,4,5][☯], Marc Molendijk[1,5][☯]*

1 Clinical Psychology Department, Leiden University, Leiden, The Netherlands, 2 Department of Psychiatry and Medical Psychology, University of Navarra, Pamplona, Spain, 3 Leiden University Treatment and Expertise Center LUBEC, Leiden, The Netherlands, 4 Department of Psychiatry, Leiden University Medical Center, Leiden, The Netherlands, 5 Leiden Institute of Brain and Cognition LIBC, Leiden University Medical Center, Leiden, The Netherlands

☯ These authors contributed equally to this work.
* f.thomas.odenthal@gmail.com (FTO); molendijkml@fsw.leidenuniv.nl (MM)

**Data Availability Statement:** All relevant data are within the manuscript and its Supporting Information files.

## Abstract

### Background

The recommendations of experts who write review articles are a critical determinant of the adaptation of new treatments by clinicians. Several types of reviews exist (narrative, systematic, meta-analytic), and some of these are more vulnerable to researcher bias than others. Recently, the interest in nutritional interventions in psychiatry has increased and many experts, who are often active researchers on this topic, have come to strong conclusions about the benefits of a healthy diet on depression. In a young and active field of study, we aimed to investigate whether the strength of an author's conclusion is associated with the type of review article they wrote.

### Methods

Systematic searches were performed in PubMed, Web of Science, Cochrane Database of Systematic Reviews, and Google Scholar for narrative reviews and systematic reviews with and without meta-analyses on the effects of diet on depression (final search date: May 30th, 2020). Conclusions were extracted from the abstract and discussion section and rated as strong, moderate, or weak by independent raters who were blind to study type. A benchmark on legitimate conclusion strength was based on a GRADE assessment of the highest level of evidence. This systematic review was registered with PROSPERO, number CRD42020141372.

### Findings

24 narrative reviews, 12 systematic reviews, and 14 meta-analyses were included. In the abstract, 33% of narrative reviews and 8% of systematic reviews came to strong conclusions, whereas no meta-analysis did. Narrative reviews were 8.94 (95% CI: 2.17, 36.84)

**Funding:** The authors received no specific funding for this work.

**Competing interests:** I have read the journal's policy and the authors of this manuscript have the following competing interests: Without relevance to this work, P. Molero reports to have received research grants from the Ministry of Education (Spain), the Government of Navarra (Spain), the Spanish Foundation of Psychiatry and Mental Health and AstraZeneca; he is a clinical consultant for MedAvanteProPhase and has received lecture honoraria from or has been a consultant for AB-Biotics, Janssen, Novumed, Roland Berger, and Scienta. This does not alter our adherence to PLOS ONE policies on sharing data and materials. The other authors declare no competing interests.

times more likely to report stronger conclusions in the abstract than systematic reviews with and without meta-analyses. These findings were similar for conclusions in the discussion section. Narrative reviews used 45.6% fewer input studies and were more likely to be written by authors with potential conflicts of interest. A study limitation is the subjective nature of the conclusion classification system despite high inter-rater agreements and its confirmation outside of the review team.

## Conclusions

We have shown that narrative reviews come to stronger conclusions about the benefits of a healthy diet on depression despite inconclusive evidence. This finding empirically underscores the importance of a systematic method for summarizing the evidence of a field of study. Journal editors may want to reconsider publishing narrative reviews before meta-analytic reviews are available.

## Introduction

New treatments do not always deliver on their promise [1]. If a new treatment that looks promising is widely adopted, the long-term effects are often less than expected from the initial evaluations [2]. This process usually takes many years, sometimes decades to complete, and it affects pharmacotherapy as well as psychotherapy [3]. Recent years have seen a marked increase of research on the effects of diet on depression. The broader field of nutrition and psychopathology has been labeled 'nutritional psychiatry' [4]. To date, a sizable amount of research exists that investigates the effects of a diet on the treatment and prevention of depression [5,6].

Treatment recommendations and guidelines should optimally rely on the evidence from multiple randomized controlled trials (RCTs), particularly RCTs that are large and well-conducted [7]. Meta-analyses and systematic reviews of RCT data provide the highest level of evidence as they synthesize and evaluate the available evidence with high standards of conduct and reporting before coming to an informed decision [8,9]. An advantage of meta-analyses over systematic reviews is that they statistically pool the available evidence and assess publication bias and between-study heterogeneity [10]. To date, one systematic review and one meta-analysis exist on the effects of dietary interventions on depressive symptoms as assessed through RCTs. The systematic review [11] presented mixed results with almost half of the studies showing a null-effect. The meta-analysis [12] showed a small positive effect of dietary interventions on depressive symptoms, but this effect is difficult to interpret because of the presence of publication bias and heterogeneity among study outcomes.

The RCTs included in the aforementioned systematic review [11] and meta-analysis [12] are typically of short duration (e.g., 8–12 weeks), and have problems with statistical power, blinding, expectation bias, and attrition [13–15]. In the event proper evidence from RCTs is absent, well-conducted prospective cohort studies may serve to provide the best available evidence [9,16,17]. The available meta-analyses of cohort studies report statistical associations between a healthy diet–in particular, the Mediterranean diet–and the incidence of depression over time [5,18]. However, reversed causation, undetected biases, and residual confounding may underlie such relationships [14,15,18,19]. Hence, these study limitations preclude strong conclusions.

Notwithstanding the limited evidence, many authors of narrative reviews come to firm conclusions about the effects of a diet on depression (e.g.: "Studies have shown that diet and nutrition play a significant role in the prevention and clinical treatment of depression" (page 10) [20]). On the contrary, authors of systematic reviews seem to come to less firm conclusions (e.g., "The results of this meta-analysis suggest that healthy pattern may decrease the risk of depression, whereas western-style may increase the risk of depression. However, more randomized controlled trails and cohort studies are urgently required to confirm this finding" (page 373) [6]). So, the clinical implementation of diet to treat or prevent depression may depend on which conclusions and recommendations are adopted by clinicians [2].

We aim to investigate whether research methods that are more sensitive to researcher bias, like narrative reviews, are more likely to overstate the benefits of a treatment–in this case, a healthy diet for depression–than research methods that are less sensitive to researcher bias, like systematic reviews with and without meta-analyses. We hypothesize that narrative reviews report more positive conclusions and recommendations about the benefits of a healthy diet on depression than systematic reviews with and without meta-analyses because narrative reviews lack the systematic method for searching and evaluating the evidence. We also hypothesize that systematic reviews without meta-analyses report more positive conclusions and recommendations than systematic reviews with meta-analyses because systematic reviews without meta-analyses do not evaluate the evidence statistically. In case different review types indeed come to different conclusions, we will explore the number of input papers, various indicators of impact, and potential conflicts of interest as possible explanations for the existence of this relationship.

## Method

As a guideline for conducting and reporting this systematic review, we followed the PRISMA [21] statement (see S1 Checklist). A protocol for this review is registered at PROSPERO, number CRD42020141372 (date of registration: April 28th, 2020).

### Search strategy

We systemically searched for meta-analyses, systematic reviews, and narrative reviews in the electronic medical databases PubMed (1964–2020), Web of Science (1974–2020), and Cochrane Database of Systematic Reviews (2005–2019) as well as the preregistration platforms PROSPERO and OSF (a free, open source web application) and the preprint servers OSF and BioRxiv from inception to May 30th, 2020. We also performed a non-systematic search in Google Scholar to identify articles that were not captured by our main search. We used the following search terms: (diet OR food) AND ("depressive*" OR depression OR "mental health" OR "mental disorder*") AND ("systematic review" OR meta-analysis OR review). We also screened the reference lists of the included articles for eligible articles. The complete search strategy is presented in S1 Text.

### Inclusion criteria

We retained peer-reviewed narrative/literature reviews, perspectives/(expert) opinions, systematic reviews, and meta-analyses on the potential effects of dietary patterns (e.g., healthy or unhealthy diet) or food groups (e.g., fruits, vegetables, or fish) on depression [22,23] and/or depressive symptoms (as measured by self-report depression symptom scales). Articles must have been written in English, German, Dutch, Spanish, or French to be included.

## Study selection

Two members of the review team (FT-O and MM) independently screened titles and abstracts of each study for eligibility. In a next round of selection, the full-text of articles was assessed for eligibility. Disagreement about the selection was resolved through consensus.

## Data extraction

FT-O and MM independently extracted the data using a prior designed extraction form. The extraction form was pilot-tested and refined accordingly. From each article, we extracted data on: a) publication date; b) number of studies included; b) whether it was a narrative review, systematic review, or meta-analysis; c) conclusions and recommendations; d) the effect sizes from meta-analyses (i.e., odds ratios, hazard ratios, relative risks, and their respective 95% confidence intervals); e) the funding sources; f) the number of input papers; g) indicators of impact (i.e., Altmetric score, impact factor of journal, number of citations); h) whether studies were written by authors with financial conflicts of interest, that is, whether authors reported to have received funding by food industry companies (e.g., Woolworth, Nestle, Taki Maki) or meat or dairy research or marketing companies (e.g., Meat and Livestock Australia); and i) whether studies were written by authors with allegiance bias. We operationalized allegiance bias as being a member of the International Society for Nutritional Psychiatry Research (ISNPR) because this society has recently published very strong conclusions about the potential role of diets on the treatment and prevention of depression [24].

From the systematic reviews with and without meta-analyses, we further extracted data on a) participant characteristics (e.g., country, total number of participants), b) method for assessing dietary patterns or food groups (e.g., diet quality scores or indexes), c) method for assessing dietary intake (e.g., food frequency questionnaires or 24-h dietary recall, food record), d) comparators (e.g., different diets or relative use), e) type of outcome (e.g., diagnosis or depressive symptoms), f) study design, and g) length of follow-up (for systematic reviews and meta-analyses of RCTs and longitudinal studies).

## Conclusion and recommendation classification

Conclusions were defined as an overall summary or interpretation of the main findings, and recommendations as an endorsement to treat or prevent depression through a diet. Conclusions and recommendations were extracted from both the abstract and discussion section. For recording the conclusions and recommendations and to reduce bias in this, we adapted the method by Antman et al [2]. One author collected all the conclusions and recommendations, while another author classified the conclusions and recommendations blind to study type into suitable categories (in this case: strong, moderate, or weak). After this, the first author categorized all the conclusions and recommendations accordingly using this classification system.

We identified *strong* conclusions through keywords, like "compelling support" or "key modifiable targets" and when authors claimed the existence of a causal relationship between diet and depression. An example of a strong conclusion would be "diet and nutrition are central determinants of mental health". *Weak* conclusions do not mention the existence of a causal relationship but, at a maximum, an association. These conclusions often come together with contrastive statements, like "however, further research is needed to establish this relationship". Keywords for weak conclusions are "suggest" or "may/might be". A *moderate* conclusion is a mix of strong and weak keywords, or weak keywords in combination with causal statements, like "diet appears to confer some protection" or "diet may prevent". A strong conclusion together with a contrastive statement also resembles a moderate conclusion. Inter-rater reliability was assessed over random selections of conclusions within the author team as

well as two neutral investigators blind to study type to further reduce potential bias. Disagreement was discussed and resolved through consensus.

## Quality assessment

We assessed the methodological quality of the included systematic reviews and meta-analyses with the AMSTAR II [25] tool. Based on each item of the quality assessment, we assessed an overall inter-rater agreement. Furthermore, we graded the certainty of the evidence in this field using the GRADE [26] approach. The certainty of the evidence reflects the extent to which we are confident that the estimate of the effect is correct–in this case, the effect of a diet on depression–and can be classified as very low, low, moderate, and high [7]. We did this to obtain a benchmark for the strength of conclusions regarding the *association* between diet and depressive symptoms as well as the *prevention* and *treatment* of depression through a healthy diet. The GRADE assessment was based on a meta-analysis on results derived from RCTs–the highest level of evidence. An evidence profile was generated [27].

## Statistical analyses

Agreement regarding the classification of conclusions was assessed through Cohen's Kappa (κ) and rank-correlation coefficients calculated over random samples of ten randomly chosen conclusions ordered from the weakest to the strongest conclusion. Ordinal regression analyses were run to test whether narrative reviews reported stronger conclusions than systematic reviews or meta-analyses. In a similar manner, associations between study type or strength of conclusions and methodological quality were assessed. In post-hoc analyses, the potential associations between study types or strength of conclusions and the number of input studies, number of citations, journal impact factor, Altmetric scores, ISNPR membership, and food industry funding were explored. To assess potential differences regarding the number of input studies and indicators of impact among the study types, independent *t*-tests were performed, or Mann-Whitney U exact tests in case the normality assumption was violated. Ordinal regression analyses were also conducted to assess putative associations between potential financial or non-financial conflicts of interest and study types or strength of conclusions. The significance level was set at an α level of 0.05, one- or two-tailed, depending on whether we tested a hypothesis or not. Odds ratios (ORs) and their respective 95% confidence intervals (CIs) were used as the measure for effect size. Analyses were performed in SPSS version 25 [28].

## Role of the funding source

There was no funding source for this study. The corresponding authors had full access to all the data in the study and had final responsibility for the decision to submit for publication.

## Results

Our initial search for meta-analyses, systematic reviews, and narrative reviews yielded 1,868 records after duplicates were removed. After reading the titles and abstracts, we excluded 1,787 records. Another 31 records were excluded after applying the inclusion and exclusion criteria (see S1 Table for the reasons for exclusion per study). Hence, we included a total number of 50 records, among which were 14 meta-analyses, 12 systematic reviews, and 24 narrative reviews/expert opinions (see Fig 1 for a flow-chart).

All systematic reviews and meta-analyses assessed the effects of a diet on depression in the healthy adult population [5,6,11,18,29–48] except for two systematic reviews that assessed children and adolescents [49,50]. The average number of participants included per systematic

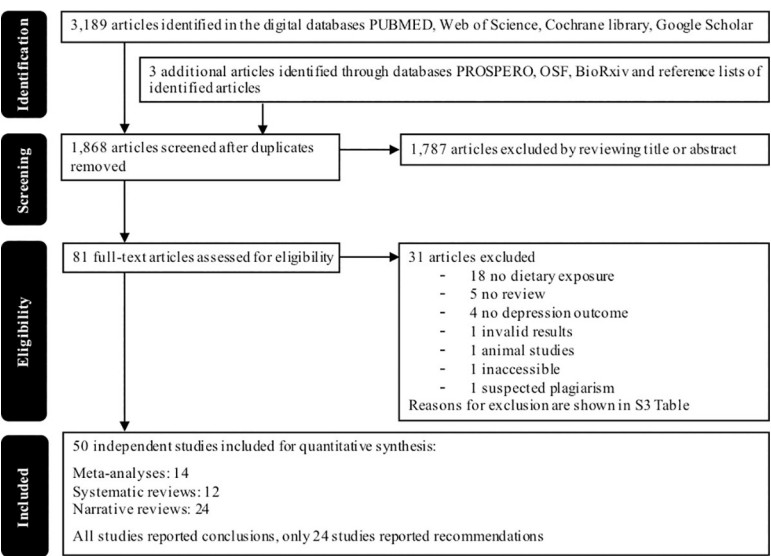

**Fig 1. Flowchart of study selection.**

review with and without meta-analyses was 128,271. In 84% of the cases, the study was performed in a Western country. The reviews mostly included prospective cohort, cross-sectional, or case-control studies, whereas five reviews investigated both observational studies and RCT [31,38–41], and one reviewed RCTs alone [11]. Most reviews included studies investigated the impact of diets, such as Mediterranean, healthy diets, or unhealthy diets. Four studies investigated the effects of fish consumption [29,33,45,46] and another four studies the effects of fruit and vegetable intake [40,42,47,48]. Food intake of participants was measured with food-frequency questionnaires (FFQs), 24-h dietary recalls, diet history questionnaires, or other standardized or non-standardized food intake questionnaires. Depression outcome was measured with standardized self-report depression scales, such as the Beck Depression Inventory or Center for Epidemiological Studies–Depression, formal diagnoses, or antidepressant medication intake. All systematic reviews and meta-analyses found positive effects of healthy diets or food groups on depression in observational studies, or negative effects of unhealthy diets or food groups. The only included systematic review of RCTs concluded that only half of the included RCTs showed significant effects for healthy diets on depression in the treatment, relative to the control group, while the other half reported null effects [11]. A full overview of the basic characteristics and results of the included meta-analyses and systematic reviews can be found in S2 Table.

We categorized the conclusions and recommendations into "strong", "moderate", and "weak" (see S3 Table) with high inter-rater agreement (average κ = 0.67, SE = 0.10, $P < 0.0001$; from four rater pairs on ten different randomly chosen papers [κ's per rater pair were 0.55, 0.63, 0.69, and 0.70]). Strong and significant average rank-correlation coefficients calculated over random samples of ten randomly chosen conclusions, ordered from the weakest to the strongest conclusion by four reviewers, further validated the reliability of our conclusion categorization (Kendall's Tau = 0.72, $P < 0.0001$ [Tau's per rater pair were 0.69, 0.78, 0.81, and 0.82]). Note that in all instances, conclusions were assessed blind to the articles from which they were derived. Furthermore, in case there was a discrepancy between assessors, these always involved differences between neighboring classifications (*e.g.*, weak *vs.* moderate and never weak *vs.* strong).

**Table 1. Strength of conclusions (abstract) per study type.**

|  | Meta-analyses | Systematic Reviews | Narrative Reviews |
|---|---|---|---|
| Strong | 0 (0%) | 1 (8.3%) | 8 (33.3%) |
| Moderate | 7 (50%) | 8 (66.7%) | 11 (45.8%) |
| Weak | 7 (50%) | 3 (25%) | 2 (8.3%) |
| None | 0 (0%) | 0 (0%) | 3 (12.5%) |

Percentages are shown in parentheses.

Tables 1 and 2 show the conclusion classification by study type of the abstract and discussion section, respectively. In the abstract, narrative reviews were 8.94 (95% CI: 2.17, 36.84) times more likely to report stronger conclusions than meta-analyses and systematic reviews, and systematic reviews were 3.43 (95% CI: 0.66 17.85) times more likely to report stronger conclusions than meta-analyses ($P = 0.001$, see Table 1). In the discussion, narrative reviews were 3.01 (95% CI: 0.95, 9.58) times more likely to report stronger conclusions than meta-analyses and systematic reviews, and systematic reviews were 2.06 (95% CI: 0.39, 10.83) times more likely to report stronger conclusions than meta-analyses ($P = 0.048$, see Table 2). After removing expert opinion papers (n = 3) from the analysis, these associations remained the same. Similarly, the associations did not change when Rahe et al [36] was treated as a systematic review instead of a meta-analysis as this study only assessed heterogeneity statistically and showed individual but not pooled effect estimates. Sensitivity analyses also showed that these patterns of results were not due to a particular study (data not shown).

Narrative reviews also appeared to report stronger recommendations in both the abstracts and discussion sections but these associations were not statistically significant ($OR_{abstract} = 1.67$, 95% CI: 0.29, 9.69, $P = 0.569$, see Table 3; $OR_{discussion} = 4.53$, 95% CI: 0.98, 20.96, $P = 0.053$, Table 4). After including the reviews into these analyses that provided no recommendations, these association became significant ($OR_{abstract} = 3.28$, 95% CI: 1.28, 8.42, $P = 0.014$; $OR_{discussion} = 3.78$, 95% CI: 1.72, 8.32, $P = 0.001$).

Inter-rater agreement regarding the AMSTAR assessment of the methodological quality of the included meta-analyses and systematic reviews was high ($\kappa = 0.91$, SE = 0.02 [~83% agreement], $P < 0.0001$; see S4 Table for details). Overall, most systematic reviews and meta-analyses were of critically low quality, except for two meta-analyses that were of low quality and one meta-analysis of moderate quality. Associations between methodological quality and conclusions and recommendations were not calculated due to a lack of variance in the quality among the study types. A more lenient quality assessment, for instance, by excluding items assessing whether studies were preregistered (item 2), whether they had language restrictions (item 4), or whether they included a list of excluded, but potentially relevant studies (item 7) did not increase variation in methodological quality to such an extent that analyses with study types were feasible.

**Table 2. Strength of conclusions (discussion) per study type.**

|  | Meta-analyses | Systematic Reviews | Narrative Reviews |
|---|---|---|---|
| Strong | 0 (0%) | 1 (8.3%) | 7 (29.2%) |
| Moderate | 9 (64.3%) | 8 (66.7%) | 12 (50%) |
| Weak | 5 (35.7%) | 3 (25%) | 5 (20.8%) |

Percentages are shown in parentheses.

**Table 3. Strength of recommendations (abstract) per study type.**

|  | Meta-analyses | Systematic Reviews | Narrative Reviews |
| --- | --- | --- | --- |
| Strong | 1 (7.1%) | 0 (0%) | 7 (29.2%) |
| Moderate | 0 (0%) | 2 (25%) | 4 (16.7%) |
| Weak | 0 (0%) | 0 (0%) | 0 (0%) |
| None | 13 (92.9%) | 9 (75%) | 13 (54.2%) |

35 out of 50 studies reported *no* recommendations in the abstract. Percentages are shown in parentheses.

ISNPR members were more likely to have written narrative reviews (OR = 5.10, 95% CI: 1.37, 19.05, $P$ = 0.015) and to have reported stronger conclusions ($OR_{abstract}$ = 4.50, 95% CI: 1.22, 16.55, $P$ = 0.024; $OR_{discussion}$ = 4.03, 95% CI: 1.10, 14.71, $P$ = 0.035) and recommendations ($OR_{abstract}$ = 18.00, 95% CI: 1.27, 255.74, $P$ = 0.033) relative to non-members (see S5 Table). ISNPR membership was also associated with a larger likelihood of having potential financial interests ($P$ < 0.01, see Table N in S5 Table). Furthermore, narrative reviews used 45.6% fewer primary input papers (mean = 11.08, standard deviation (SD) = 6.98) than systematic reviews and meta-analyses (mean = 20.35, SD = 12.18; $P$ = 0.002). No statistically significant associations existed between industry funding and study types (OR = 2.43, 95% CI: 0.54, 10.87, $P$ = 0.246) or strength of conclusions ($OR_{abstract}$ = 3.01, 95% CI: 0.65, 13.95, $P$ = 0.158; $OR_{discussion}$ = 3.60, 95% CI: 0.74, 17.48, $P$ = 0.112). There were no significant differences in number of citations, Altmetric scores, and journal impact factors as a function of study type.

All associations remained similar after controlling for publication year, number of input papers, or journal impact factor.

## GRADE assessment of meta-analysis of RCTs

The GRADE approach was applied to obtain a benchmark for the strength of conclusions regarding the association between diet and depressive symptoms. Initially, we wanted to apply the GRADE assessment on Firth et al [12] as they present the only meta-analysis of studies reporting on the effects of dietary interventions on depressive symptoms derived by means of RCTs. However, we noted two crucial errors in the article that, when corrected, would lead to a substantially different result. An erratum for this study has already been published [51]; however, we think the meta-analysis has not been corrected to a sufficient degree. We, therefore, decided to rerun this meta-analysis following the exact same approach of Firth et al [12] to obtain corrected results as well as the input material for the GRADE assessment: One notable difference relative to Firth et al [12] is that we have applied an extended final search date (August 3rd, 2019 *versus* December 3rd, 2018). Another difference is that we did not pool all the data in one analysis. Instead, we analyzed the data separately to answer the following questions: 1) can a healthy diet *prevent* depression? 2) can a healthy diet *treat* depression? and 3) is

**Table 4. Strength of recommendations (discussion) per study type.**

|  | Meta-analyses | Systematic Reviews | Narrative Reviews |
| --- | --- | --- | --- |
| Strong | 1 (7.1%) | 2 (16.7%) | 14 (58.3%) |
| Moderate | 1 (7.1%) | 4 (33.3%) | 2 (8.3%) |
| Weak | 0 (0%) | 0 (0%) | 0 (0%) |
| None | 12 (85.7%) | 6 (50%) | 8 (33.3%) |

26 out of 50 studies reported *no* recommendations in the discussion. Percentages are shown in parentheses.

**Table 5. GRADE summary-findings table based on a newly performed meta-analysis of RCTs.**

| Outcome | Effect size (95% CI) | k | N intervention / control | Certainty of evidence |
|---|---|---|---|---|
| Prevention | g = 0.06 (-0.10, 0.22) | 2 | 512 / 513 | Low (⊕⊕◯◯) [a] |
| Treatment | g = -0.27 (-0.66, 0.13) | 4 | 115 / 101 | Very low (⊕◯◯◯) [b] |
| Association | g = -0.14 (-0.24, -0.04) | 15 | 18,622 / 26,877 | Very low (⊕◯◯◯) [c] |

*Abbreviations.* CI, confidence interval; *g*, Hedges' g; *N*, number of participants.

[a] Downgraded once for serious risk of bias, once for imprecision.

[b] Downgraded twice for very serious risk of bias, once for imprecision.

[c] Downgraded twice for very serious risk of bias, once for inconsistency.

a healthy diet *associated* with a reduction in depressive symptoms over time? This was done to reduce between-study heterogeneity and because of the GRADE requirement that questions are formulated regarding specific populations, interventions, comparators, and outcomes.

The meta-analyses revealed no evidence for the hypotheses that a diet can treat or prevent depression. A small statistically significant benefit of a healthy diet on depressive symptoms was found in association studies that did not specifically aim to prevent or treat depression (see Table 5). Yet, substantial and significant between-study heterogeneity ($I^2$ = 49%) was observed. Cumulative meta-analysis also showed that in 2000, 2005, 2010, and 2015 there was no ground to formulate strong conclusions regarding an effect of a diet on depression. For further information on this meta-analysis, we refer to S2 Text.

A full overview of the GRADE evaluation and the reasons for up- or downgrading the evidence can be found in S1 Appendix. In sum, GRADE indicated *very low* to *low* certainty-evidence for the proposed associations between diet and depression that are under study here (see Table 5). These findings, thus, indicate that strong conclusions about the potential effects of diet on depression are not warranted.

## Discussion

This systematic review reports substantial discrepancies in the strength of conclusions reported over study types. Narrative reviews were more likely to report stronger conclusions and recommendations regarding the benefits of a healthy diet on depression relative to systematic reviews and meta-analyses, whereas systematic reviews were slightly more likely to report stronger conclusions and recommendations than meta-analyses. In fact, no single meta-analysis came to a strong conclusion regarding the supposed effect of diet on depression. In line with this was the result of a GRADE evaluation of the highest level of evidence, which dictated that the certainty of the evidence is low regarding the prevention, and very low regarding the treatment of depression through a healthy diet as well as the association between diet and depressive symptoms over time. An AMSTAR assessment revealed that the methodological quality of meta-analyses and systematic reviews was mainly critically low. Hence, we can conclude that a substantial part of narrative reviews, and a minor part of systematic reviews, overstate the benefits of a healthy diet on depression.

Although we can only speculate on explanations underlying the biased conclusion formulation, we did find some informative correlates of it. Narrative reviews used 45.6% fewer input studies. This finding could indicate that authors selectively cited easily accessible or mainly positive input material [52]. Selective citation practices could be justified if authors who came to strong conclusions would cite the *scientifically strongest articles* only. Yet, this seems unlikely since our GRADE assessment showed that no such strong articles exist because of the presence of serious risk of bias in the study designs and executions. Additionally, our meta-analyses of

RCTs indicated that there is no evidence for the notion that a diet can treat depression or prevent it from occurring. Financial and non-financial conflicts of interest may also affect narrative reviews more than other types of reviews. ISNPR members wrote 30% of all papers, 45.8% of narrative reviews, and only two meta-analyses [5,12] on this topic. One of these meta-analyses [5] was included in this systematic review, while the other [12] was excluded because of crucial errors favoring the authors' study hypotheses. As in all areas of health research, many factors may underlie the formulation of biased conclusions, ranging from a drive to make a positive contribution to personal experiences to financial interests [53–56].

Food industry-funded studies may also more likely to report stronger conclusions than non-funded studies. The effect size of this association was large (OR = 3.60), but it did not reach statistical significance. Authors tend to underreport their financial conflicts of interest [57], but we do not know if this is also the case in nutritional mental health research. Yet, a recent debate about a new dietary recommendation on red meat consumption disclosed that financial conflicts of interest may indeed play a role also in the nutritional field [58].

Authors of narrative reviews tend to report stronger conclusions in the abstracts than in the discussion sections among the study types. Word limits of abstracts may force authors to generate more generic conclusions [59]. As most readers may only read the title and abstract, we think the abstract should already convey the right message to begin with. When scientific experts conclude that "diet and nutrition are central determinants of mental health" and that "nutrition is a crucial factor in the high prevalence and incidence of mental disorders" (page 271) [24], the data that underlie these conclusions should be convincing. This is not yet the case. As a consequence, the general media have disseminated grossly overstated conclusions [60,61]. A healthy diet has few downsides, but patients may inaccurately believe that they are themselves responsible for their depression (*e.g.*, "My bad dietary habits made me depressed").

A strength of this systematic review is that we used a transparent and quantitative method for identifying study limitations and their potential sources of bias. We applied a systematic method for searching and extracting relevant data with high inter-rater agreements. We used rigorous, scientific, and standardized instruments to assess the certainty of the evidence (GRADE) and the methodological quality (AMSTAR) to gauge which conclusions are most likely correct. Furthermore, we obtained the highest level of evidence regarding the effects of diet on depression and reduced between-study heterogeneity by pooling, through a meta-analysis, results from RCTs based on their populations, interventions, comparators, and outcomes (*i.e.*, prevention, treatment, association), which increases the validity of our findings [62].

A limitation of our work is the subjective nature of our conclusion classification system, although this was done with high inter-rater agreements, which was confirmed outside of the review team. Secondly, the GRADE approach may be less applicable for lifestyle interventions as the evidence from large and well-conducted RCTs is often absent; more lenient rules for appraising the evidence have been suggested (e.g., HEALM [63]). Thirdly, the present research is limited to the field of diet and depression but similar inferences may apply to different exposures, like nutraceuticals, and different outcomes, like cardiovascular health (our GRADE evaluation of other patient-relevant health outcomes already indicated low to very low certainty-evidence also in these fields of study; see S1 Appendix). Lastly, there may be sub-samples in the population for whom diet may directly affect mood (*e.g.*, people with celiac disease) [64]. However, this was hardly, if ever, acknowledged in the generic conclusions that we encountered and was, therefore, also not investigated here. Future research should investigate all this and invest in large-scale and long-term randomized controlled dietary intervention and prevention trials.

Our work indicates that conclusions derived from systematic reviews with meta-analyses should be favored over those derived from less rigorous scientific methods, like narrative

reviews. Awareness of this should be high on the agenda of journal editors and reviewers to reconsider publishing narrative reviews before meta-analytic reviews are available. Our work may also encourage researchers to use systematic reviews instead of narrative reviews to protect themselves against their own biases. The preregistration and open access of such work may further reduce these researcher biases.

## Supporting information

**S1 Checklist. PRISMA checklist.**
(DOC)

**S1 Text. Search strategy.**
(DOCX)

**S2 Text. Details on our meta-analysis of RCTs.**
(DOCX)

**S1 Table. Excluded studies with reason.**
(DOCX)

**S2 Table. Basic characteristics and results of assessed systematic reviews.**
(DOCX)

**S3 Table. Reported conclusions and recommendations of included systematic reviews with classified strength.**
(DOCX)

**S4 Table. AMSTAR II scoring.**
(DOCX)

**S5 Table. Association between study types and strength of conclusion, number of input papers, indicators of impact, and potential conflicts of interest.**
(DOCX)

**S1 Appendix. GRADE evaluation of three patient-relevant healthcare questions.**
(DOCX)

## Acknowledgments

We thank our colleagues from the Universities of Leiden and Navarra for the discussions about the topic of this manuscript on formal and informal occasions. We also want to thank Anouk Mentink, Cristina Vidal Adroher, Liv Caro Henrich, Marta Santos Burguete, and Mirjam Christina Reidick for their help in classifying conclusions and/or proof-reading of earlier versions of his manuscript.

## Author Contributions

**Conceptualization:** Florian Thomas-Odenthal, Patricio Molero, Willem van der Does, Marc Molendijk.

**Data curation:** Florian Thomas-Odenthal, Marc Molendijk.

**Formal analysis:** Florian Thomas-Odenthal, Marc Molendijk.

**Investigation:** Florian Thomas-Odenthal, Patricio Molero, Marc Molendijk.

**Methodology:** Florian Thomas-Odenthal, Patricio Molero, Willem van der Does, Marc Molendijk.

**Project administration:** Marc Molendijk.

**Resources:** Willem van der Does, Marc Molendijk.

**Software:** Florian Thomas-Odenthal, Marc Molendijk.

**Supervision:** Marc Molendijk.

**Validation:** Florian Thomas-Odenthal, Patricio Molero, Willem van der Does, Marc Molendijk.

**Visualization:** Florian Thomas-Odenthal, Willem van der Does, Marc Molendijk.

**Writing – original draft:** Florian Thomas-Odenthal.

**Writing – review & editing:** Florian Thomas-Odenthal, Patricio Molero, Willem van der Does, Marc Molendijk.

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
