## [Decision Letter · Decision Letter 0]

19 May 2020

PONE-D-20-10335

Diets against depression: strong conclusions, weak evidence. A systematic review

PLOS ONE

Dear Mr Thomas-Odenthal,

Thank you for submitting your manuscript to PLOS ONE. After careful consideration, we feel that it has merit but does not fully meet PLOS ONE’s publication criteria as it currently stands. Therefore, we invite you to submit a revised version of the manuscript that addresses the points raised during the review process.

Please follow the instructions of both reviewers, particularly with regard to clarifying the objectives and the presentation of the results. You have received specific instructions on the searching, and if there are actions that you are unable to take with regard to the searches please include this in the discussion as a limitation. 

We would appreciate receiving your revised manuscript by Jul 03 2020 11:59PM. To enhance the reproducibility of your results, we recommend that if applicable you deposit your laboratory protocols in protocols.io, where a protocol can be assigned its own identifier (DOI) such that it can be cited independently in the future. For instructions see: http://journals.plos.org/plosone/s/submission-guidelines#loc-laboratory-protocols

We look forward to receiving your revised manuscript.

Kind regards,

Lisa Susan Wieland

Academic Editor

PLOS ONE

Journal Requirements:

2. Please modify the title to ensure that it is meeting PLOS’ guidelines (https://journals.plos.org/plosone/s/submission-guidelines#loc-title).

In particular, the title should be "specific, descriptive, concise, and comprehensible to readers outside the field" and in this case it is not informative and specific about your study's scope and methodology.

Please ensure that you amend both the title on the online submission form (via Edit Submission) and the title in the manuscript so that they are identical.

'I have read the journal's policy and the authors of this manuscript have the following competing interests:

Without relevance to this work, P. Molero reports to have received research grants from the Ministry of Education (Spain), the Government of Navarra (Spain), the Spanish Foundation of Psychiatry and Mental Health and AstraZeneca; he is a clinical consultant for MedAvanteProPhase and has received lecture honoraria from or has been a consultant for AB-Biotics, Janssen, Novumed, Roland Berger, and Scienta. The other authors declare no competing interests.'

Reviewers' comments:

Reviewer's Responses to Questions

**Comments to the Author**

1. Is the manuscript technically sound, and do the data support the conclusions?

Reviewer #1: Partly

Reviewer #2: Partly

2. Has the statistical analysis been performed appropriately and rigorously? 

Reviewer #1: Yes

Reviewer #2: I Don't Know

3. Have the authors made all data underlying the findings in their manuscript fully available?

Reviewer #1: Yes

Reviewer #2: Yes

4. Is the manuscript presented in an intelligible fashion and written in standard English?

Reviewer #1: Yes

Reviewer #2: Yes

5. Review Comments to the Author

Reviewer #1: This is a very interesting manuscript. I have some suggestions that may help improve some aspects of the manuscript:

- General: some sentences need revision in order to be correct (e.g. past /present tenses; and some words have missing letters).

- One of the main aspects which need to be clarified in the abstract and in the main manuscript is the exact objective (see comment below).

- Abstract: I suggest to be more precise in the objective of the work presented and revise which aspects are important to be mentioned in the abstract (and which are not so important). I think that the abstract needs more structure to present the main focus of the manuscript.

- Introduction: Authors should be sure that all statements are referenced (e.g. first 4 sentences of the introduction need to be references).

- If I understood correctly, the objective of the present review is to investigate the difference between the conclusions given in the different type of reviews (narrative reviews, systematic reviews (with/and without meta-analysis)) regarding diet and depression. In this sense, I would suggest authors to rephrase the introduction, in order to backup their objective. For example, they could mention which kind of literature is found for diet and depression; give an overview (and definition) of narrative review vs. systematic review (with/without meta-analysis); and how conclusion of each type of review could influence practice.

- Line 52-55: Recommendations and guidelines DO NOT HAVE TO rely on RCTs. It is correct, that the best evidence usually stems from RCTs, because they can investigate causality (not because standards of conduct are high: there are RCTs with low quality standards, and also non-RCTs with high quality standards). Also, there are some research questions which cannot be answered with RCTs (which is often the case in nutritional topics), thus guidelines have to be done with non RCTs. Suggest authors to revise some literature about evidence and study designs and rephrase this section.

- Recommend to revise the following terms used: systematic review and meta-analysis. They are not strictly different type of reviews. A systematic review can include a meta-analysis, but does not necessary have to include one. My recommendation would be to use: narrative reviews, systematic reviews without meta-analysis, and systematic reviews with meta-analysis.

- Method: Suggest to include the date of registration in PROSPERO (also in the Supl. material where the registration form is presented)

- Method / Search strategy: please explain what OSF means.

- Why did authors restrict your search from January 2019 until September 2019?

- Did authors use only free-text search or also controlled vocabulary?

- In the PROSPERO registration form, authors list more databases that were searched. Which databases were included in the final version of the review?

- Suggest to include the whole search strategy as Supl. material, since the search strategy presented seems incomplete (also when opening the link provided in the PROSPERO form, the search strategy is incomplete).

- Method / Inclusion criteria: which depressive symptoms were considered eligible?

- Method / Data extraction: how did authors address missing data?

- Method / Conclusion and recommendation classification: how did authors define the conclusion, in cases where the study reported different statements between the conclusion in the abstract section and in the main manuscript? Also, one study could report on various conclusions: how was this handled (where all conclusions extracted and rated)?

- Lines 149-154: I did not understand what this paragraph is describing about the method of GRADE? To what outcomes did authors assess GRADE? Did they do this to all included reviews or only to the identified meta-analyses?

- Method / Statistical analysis: did authors assess association with correlation coefficient (Spearman, Pearson)? Please specify.

- Results: Suggest to start the results section more general. How many references were identified in the databases, how many screened, how many included, etc. Before presenting any results it is important to give the reader an overview about the studies/publications included, their general characteristics, etc.

- Authors start the results describing their GRADE assessment of one meta-analysis. I do not understand how this should be understood contextualizing their objective? I think authors should first give results addressing their objective (i.e. how many narrative reviews, systematic reviews, etc. were found and what did they conclude).

- Lines 196-200: which outcomes were assessed with GRADE? The manuscript is lacking an overview about which publications were taken into consideration for the GRADE approach (this is also missing in the methods section).

- Line 202: I think this is how the whole results section should start with.

- Line 205: the study of Firth et al was excluded, but their data was used to re-run meta-analysis (described at the beginning of the results section)? This is confusing.

- Suggest to re-order the results section and provide all necessary information for the reader in order to understand the results. Supplementary material is vast and includes all results, but I suggest to depict the most important results also in the text.

- The conclusion sections is well written and has all elements needed in the conclusion section. As an option, authors could include some further limitations about the design of their investigation (e.g. search strategy)?

Reviewer #2: Methods

Search strategy

1. although the authors state they used PRISMA to report their study, the description of the search strategy is not compliant with that guideline. The authors don’t report the coverage of the databases – the period of time covered in the database when it is searched (not the period of time during which the searches were run by the investigators). More importantly, it seems unlikely the search strategy reported is the actual search run in each database (PRISMA recommends reporting of at least one search strategy “…such that it could be repeated”), because it lacks database-specific syntax or indication of use of field codes.

2. If this is the actual search strategy, it is not systematic or comprehensive:

a) The lack of field codes suggest that, at best, the authors conducted a basic search in PubMed, while not taking advantage of the extensive controlled vocabulary (Medical Subject Headings) that should be used to ensure thorough searching.

b) The search terms used do not appear to fully reflect “diet” as conceptualized by the authors. That is, the authors wanted to include reviews that examined the impact of consumption of food groups such as fruits, vegs, or fish on depressive symptoms. Yet the search string contains no terms about food groups.

c) At the same time, several search terms seem inappropriate for the topic of depression: anxiety, psychosis.

3. The protocol for this study indicates that Cochrane Library and Google Scholar would also be searched, but this does not appear to be the case? Please clarify if these were completed and if so, why the results are not included here. If not done, please explain why.

Inclusion criteria

Expert opinion articles do not seem to me to be in the same category of document as reviews or meta analyses, so it’s not clear why they should have been included in this study. Given their nature and typically brief format, it is expected that such documents would be selective in the studies cited and perhaps adamant in the conclusions drawn, so lumping them in with narrative reviews (which is done in the Results) might lead to bias.

Conclusion and Recommendation Classification

If conclusions and recommendations were provided in both the abstract and discussion section of a review, the authors extracted only the information in the abstract. I’d like to see a rationale for doing this. Wouldn’t it be more appropriate to extract from both sections, and to look at the impact of section on the outcomes the authors are interested in?

Results

The entire first paragraph here (lines 174-187) seems to be about the meta analysis the authors used to create what I will call a “gold standard” for the true or real effect of diet on depressive symptoms. I think the manuscript would flow better if this paragraph were in the Methods section.

Association between study type and strength of conclusions

1. I’m concerned about the data as reported here and in the flow chart (Fig 1).

a) minor detail: OSF and PROSPERO are not bibliographic databases, so the # of records found by those searches should be included in the box used to indicate “additional articles identified through other means”.

b) The initial search results, across all databases and resources, found a total of 1696 records. The flow chart indicates that, after duplicate removal, 1696 records remained. So there were no duplicates? I work in an environment where colleagues are collaborating on multiple SRs per year, and I have never heard of a systematic review where the searches produced no duplicates. That oddity needs to be explained

6. PLOS authors have the option to publish the peer review history of their article (what does this mean?). If published, this will include your full peer review and any attached files.

Reviewer #1: Yes: Daniela Küllenberg de Gaudry

Reviewer #2: No

---

## [Author Response · Author response to Decision Letter 0]

3 Jul 2020

We thank Lisa Susan Wieland, the academic editor of PLOS ONE, for the opportunity to submit a revised version of our manuscript. We also thank the reviewers for their valuable feedback. We incorporated the feedback and replied to each comment by the reviewers individually in the "response to reviewers" document. Furthermore, we clarified the objectives by focusing more on the association between conclusion strength and review types instead on the researcher biases underlying such conclusions. We made this clearer specifically in the abstract and introduction. We also altered the presentation of the results by first presenting the results of our systematic review while replacing the section on the GRADE approach to the end. We think this way the reading flow of the paper has improved without losing relevant elements. Moreover, we improved our search strategy based on the reviewers’ comments and conducted an entirely new search. We found 9 additional papers and this time we extracted conclusions from the abstract as well as the discussion. Based on this data, we performed separate analyses on the associations between conclusions of abstracts and discussion sections with review types with similar results. Lastly, we adapted our manuscript to the style requirements of the journal, altered the title, and updated our competing interest section by including data availability statements all in accordance with PLOS ONE guidelines. The competing interest statement itself has not altered. By incorporating the feedback, we think the paper has become much more valuable and thereby interesting for the readership of PLOS ONE. We hope the editors and reviewers agree.

---

## [Decision Letter · Decision Letter 1]

11 Aug 2020

Impact of review method on the conclusions of clinical reviews: A systematic review on dietary interventions in depression as a case in point.

PONE-D-20-10335R1

Dear Dr. Thomas-Odenthal,

We’re pleased to inform you that your manuscript has been judged scientifically suitable for publication and will be formally accepted for publication once it meets all outstanding technical requirements.

Kind regards,

Lisa Susan Wieland

Academic Editor

PLOS ONE

Additional Editor Comments (optional):

There is one additional item that I would like you to consider amending, however it is optional. The plain language equivalent of low certainty evidence in GRADE should use the term 'may' to reflect the low level of certainty. Therefore, in S.9 Appendix when translating low certainty evidence, it is better to say ‘The evidence suggests that a healthy diet may not prevent depressive symptoms.’ instead of ‘The evidence suggests that a healthy diet does not prevent depressive symptoms.’ Consider changing this and the other plain language statements for low quality evidence. 

Reviewers' comments:

Reviewer's Responses to Questions

**Comments to the Author**

1. If the authors have adequately addressed your comments raised in a previous round of review and you feel that this manuscript is now acceptable for publication, you may indicate that here to bypass the “Comments to the Author” section, enter your conflict of interest statement in the “Confidential to Editor” section, and submit your "Accept" recommendation.

Reviewer #1: All comments have been addressed

Reviewer #2: All comments have been addressed

2. Is the manuscript technically sound, and do the data support the conclusions?

Reviewer #1: Yes

Reviewer #2: Yes

3. Has the statistical analysis been performed appropriately and rigorously? 

Reviewer #1: I Don't Know

Reviewer #2: I Don't Know

4. Have the authors made all data underlying the findings in their manuscript fully available?

Reviewer #1: Yes

Reviewer #2: Yes

5. Is the manuscript presented in an intelligible fashion and written in standard English?

Reviewer #1: Yes

Reviewer #2: Yes

6. Review Comments to the Author

Reviewer #1: (No Response)

Reviewer #2: The authors have done a good job of responding to my comments. I have one minor suggestion: they indicate they searches "Cochrane reviews". I believe they are referring to the Cochrane Database of Systematic Reviews. if so, that name should replace "cochrane reviews" throughout the manuscript.

7. PLOS authors have the option to publish the peer review history of their article (what does this mean?). If published, this will include your full peer review and any attached files.

Reviewer #1: No

Reviewer #2: No

---

## [Editor Report · Acceptance letter]

17 Aug 2020

PONE-D-20-10335R1 

Impact of review method on the conclusions of clinical reviews: A systematic review on dietary interventions in depression as a case in point. 

Dear Dr. Thomas-Odenthal:

I'm pleased to inform you that your manuscript has been deemed suitable for publication in PLOS ONE. Congratulations! Your manuscript is now with our production department. 

Kind regards, 

on behalf of

Dr. Lisa Susan Wieland 

Academic Editor

PLOS ONE